# Image Translation of Breast Ultrasound to Pseudo Anatomical Display by CycleGAN

**DOI:** 10.3390/bioengineering10030388

**Published:** 2023-03-22

**Authors:** Lilach Barkat, Moti Freiman, Haim Azhari

**Affiliations:** Biomedical Engineering Faculty, Technion-Israel Institute of Technology, Haifa 3200001, Israel; lilachbarkat@campus.technion.ac.il (L.B.); moti.freiman@bm.technion.ac.il (M.F.)

**Keywords:** ultrasound, image translation, CycleGAN, breast tumors

## Abstract

Ultrasound imaging is cost effective, radiation-free, portable, and implemented routinely in clinical procedures. Nonetheless, image quality is characterized by a granulated appearance, a poor SNR, and speckle noise. Specific for breast tumors, the margins are commonly blurred and indistinct. Thus, there is a need for improving ultrasound image quality. We hypothesize that this can be achieved by translation into a more realistic display which mimics a pseudo anatomical cut through the tissue, using a cycle generative adversarial network (CycleGAN). In order to train CycleGAN for this translation, two datasets were used, “Breast Ultrasound Images” (BUSI) and a set of optical images of poultry breast tissues. The generated pseudo anatomical images provide improved visual discrimination of the lesions through clearer border definition and pronounced contrast. In order to evaluate the preservation of the anatomical features, the lesions in both datasets were segmented and compared. This comparison yielded median dice scores of 0.91 and 0.70; median center errors of 0.58% and 3.27%; and median area errors of 0.40% and 4.34% for the benign and malignancies, respectively. In conclusion, generated pseudo anatomical images provide a more intuitive display, enhance tissue anatomy, and preserve tumor geometry; and can potentially improve diagnoses and clinical outcomes.

## 1. Introduction

Breast cancer is the second most prevalent cancer among women in the United States and the second leading cause of cancer death among women overall. Early diagnosis through screening, alongside treatment advancements, have substantially reduced breast cancer mortality [1]. The screening and diagnosis process is primarily conducted using mammography, ultrasound, and dynamic contrast-enhanced magnetic resonance imaging (DCE-MRI). Each of these modalities has its own unique advantages and uses, depending on the patient’s age, breast tissue density, medical history, and the tumor’s stage and type [2].

Ultrasound is a major tool in breast cancer diagnosis. It is probably the most cost-effective medical imaging modality available today. Stemming from its high availability and from the fact that it is considered hazardless, it has found applications in almost all fields of medicine. Nonetheless, even though it was introduced to medicine about eighty years ago and despite the enormous progress in the fields of electronics and computers, one factor remained unchanged: ultrasound images are characterized by a granulated appearance with poor a SNR and substantial speckle noise. Furthermore, image quality is subject to the skills of the scanner operator and the manner in which the probe is attached and pressed against the tissue [3]. As a result, anatomical visualization is relatively inferior when compared to mammography or MRI. Consequently, the clinical information is compromised, and image interpretation relies substantially on the skills of the practicing radiologist [4,5].

On the positive side, since ultrasound image acquisition rate enables real-time imaging, ultrasound is commonly used during surgery to bridge the gap between the preliminary radiological images acquired before surgery and the actual area of tissue operated on [6,7,8]. Still, there is often a challenge in identifying the corresponding pathology of interest in the anatomical cut. The difficulty stems mainly from two reasons. The first problem is the potential mismatch between the spatial orientation of the image acquired before surgery relative to the actual cut and the physical position of the patient at the time of the image acquisition. Relating to this problem, ultrasound offers an advantage over other imaging modalities. Since ultrasound scans are conducted by a handheld probe, they can yield arbitrary cross-sectional images from various directions in real time. This may enable the surgeon to find the best match with the preoperative radiological information during the surgery. The second challenge is the substantial differences in appearance between the black and white radiological image and the texture and color of the operated tissue. The remedy for that could be a style of display which resembles the anatomical appearance more closely.

Another significant challenge faced during surgery is ensuring the complete removal of all the tumor tissue(s). This is typically achieved by cutting a margin of healthy tissue surrounding the tumor. The remaining positive margin, which refers to residual tumor tissue that remains in the body after the initial surgical resection, is a very important factor that determines the success of the surgery. Intraoperative margin assessments encompass a range of techniques aimed at providing information on the composition of the tissue and discriminating between normal and malignant tissue during an operation. One such approach is spectroscopy, which evaluates biochemical differences by analyzing the aerosol released during electrosurgery [9] or by measuring the frequency shift of light scattered by a focused laser [10]. Another method involves the determination of tissues’ electrical properties through measurement of their resistance [11]. Molecular imaging, which entails the detection of specific contrast agents or dyes, is also utilized for intraoperative margin assessment [12,13]. Following the surgical procedure, the evaluation of margins is customarily carried out through histopathology [14] to examine the excised tissue and guarantee the presence of adequate healthy margins.

Intraoperative ultrasound offers another option. It enables mapping the margins around the tissue in situ and examining the excised tissue. Detection of positive margins in situ through the use of ultrasound enables the surgeon to make informed decisions regarding additional resection procedures in real-time, if needed [6]. In addition, the ultrasound probe can be attached directly to the excised tissue immediately after removal from the breast, and the displayed images can be used for measuring the sizes and locations of the healthy margins. This enables real-time margin assessment within the operation room. Thus, if needed, more tissue can be removed. This can prevent cases where close or positive margins are identified post-surgery, which mostly leads to additional surgeries, treatment delay, and increase patient anxiety [15]. In this context as well, a display that resembles an actual anatomical cut can be advantageous.

The aim of this study is harnessing the numerous advantages of ultrasound imaging that were mentioned, both pre- and intra-operatively, while addressing its primary drawbacks in relation to other imaging modalities, which are the poor quality and interpretability of the resultant images. To bridge this gap, our research focused on generating a more intuitive representation of ultrasound images that would facilitate their interpretation by the operators while preserving their anatomical information. An overview of the advantages and disadvantages of ultrasound imaging is presented in Table 1.

In recent years, attempts have been made to generate methods for medical images’ translation into a more convenient form. For example, take translating point-of-care ultrasound to higher-quality ultrasound [16], translation between MRI different protocols [17], and cross modality translations, such as ultrasound to MRI [18] and MRI to CT [19]. Further, ultrasound image visualization has been improved by using CT as a reference [20]. Image translation from different domains can be done by implementing one of several methods [21], most of which are based on generative adversarial network (GANs) [22] or on variational auto encoders (VAEs) [23]. The variety of methods can broadly be categorized into supervised and unsupervised methods. Supervised methods require paired images for training, whereas unsupervised methods do not [21]. One optional translation method is the cycle generative adversarial network (CycleGAN). This unsupervised method is capable of learning the mapping between two image domains without the need for paired training data.

In this work, we aimed to explore a different approach by creating a novel visual representation of the tumor shape. It is postulated here that a transformation into a more realistic and more intuitive display will be beneficial, by allowing a more straightforward interpretation. We have chosen to transform the ultrasonic images into a pseudo anatomical display, as a demonstration of feasibility, which seems to be more natural to comprehend. This form of representation, with colors and clearer borders, has the potential to be more intuitive for the surgeon, and especially for those who are less experienced in the field. It could potentially simplify ultrasound findings interpretation and improve the surgical procedure. Since in our case, it was physically impossible to produce perfectly matched image pairs for training the network, we hypothesized that this can be achieved by using CycleGAN.

## 2. Materials and Methods

### 2.1. Cycle Generative Adversarial Network (CycleGAN)

In order to create a transformation between two image domains using AI, it is customary to establish a large number of paired images to train the network. This requires that each pair of images would be acquired under the same exact conditions and views. Evidently, due to the operator-dependent manner of ultrasonic-image acquisition, it may be impractical to obtain matching pairs of images that exactly correspond to the same anatomical cross-sections. However, the introduction of CycleGAN [24] has laid the foundations for a new approach. CycleGAN, which has already been applied in several studies in the medical field [18,25,26], is used to study the features of each image domain separately and enforce similarity in the cross-domain. This, therefore, enables us to overcome the lack of paired-image datasets (i.e., ultrasound vs. optical/anatomical). Additionally, another notable advantage of CycleGAN is its translation rapidity. Following the training of the neural network, the inference time is a fraction of a second, and thereby the images can be translated and displayed alongside the ultrasound images in real time.

The CycleGAN architecture is comprised of two generative adversarial network (GAN) units, one for each image domain. As shown in Figure 1, each of the GAN models has a generator and a discriminator, which are trained simultaneously. The generator attempts to produce from the input image domain a realistic display which fits the target domain as well as possible. The discriminator, on the other hand, examines the features of the produced image and decides whether or not it belongs to the target domain. According to the discriminator feedback, the generator tries to improve the generated image. In parallel, the discriminator accepts actual input images that belong to the target domain and improve its discrimination capability. Thus, the two networks enforce mutual improvement in an adversarial manner.

In the context of this work, one generator was trained to produce pseudo anatomical displays ($G_{PA}$) from ultrasonic images, and its corresponding discriminator ($D_{PA}$) was trained to distinguish real from generated synthetic images, as shown in Figure 1a. In an alternating fashion, the second generator was trained to produce ultrasound displays ($G_{US}$) from optical anatomical images, and simultaneously trained its corresponding discriminator ($D_{US}$) (see Figure 1b).

Referring to mathematical terms, the real ultrasonic image is defined as *x* and the real pseudo anatomical image as *y*. Accordingly, the adversarial loss for each of the GAN units is calculated by [24]:(1)$$\begin{array}{cc} L_{GAN_{PA}}\left(G_{PA},D_{PA}\right)= & E_{y∼pdata(y)}\left[log\left(D_{PA}\left(y\right)\right)\right]+\\ \; & E_{x∼pdata(x)}[log\left(1-D_{PA}\left(G_{PA}\left(x\right)\right)\right] \end{array}$$
(2)$$\begin{array}{cc} L_{GAN_{US}}\left(G_{US},D_{US}\right)= & E_{y∼pdata(y)}\left[log\left(D_{US}\left(y\right)\right)\right]+\\ \; & E_{x∼pdata(x)}[log\left(1-D_{US}\left(G_{US}\left(x\right)\right)\right] \end{array}$$
where $L_{GAN_{PA}}$ and $L_{GAN_{US}}$ are the loss functions for ultrasound and PA, respectively, and $E_{x∼pdata}$ and $E_{y∼pdata}$ are the corresponding expected values.

In addition, the networks were trained to maintain cycle-consistent translation, which means minimizing the difference between the original image and its reconstruction. In order to enforce this similarity in the cross domain, the input image was passed through one generator, followed by a second generator, minimizing the discrepancy of the real input image from the reconstructed image.

Within this framework, the cycle consistency is determined by the translation of ultrasonic images to anatomical displays and back to ultrasonic images, or from anatomical images to ultrasonic displays and back. These cycle-consistent translations are depicted visually by the green arrows in Figure 1. Maintaining cycle consistency is important, as it ensures that the networks are able to preserve the original information and details of the images during the translation process. The cycle reconstruction loss $\left(L_{cycle}\right)$ was calculated by:(3)$$\begin{array}{cc} L_{cycle}\left(G_{PA},G_{US}\right)= & E_{x∼pdata(x)}\left[|\right|G_{US}\left(G_{PA}\left(x\right)\right)-{x||}_{1}]+\\ \; & E_{y∼pdata(y)}\left[|\right|G_{PA}\left(G_{US}\left(y\right)\right)-{y||}_{1}] \end{array}$$

In the original paper of CycleGAN [24], the researchers proposed the usage of an identity loss to preserve the pixels’ colors throughout the image translation. However, in our case, the two domains differ substantially in their color scales and contrast properties. While the tumors have a relatively white color to the human eye, they appear as black regions in ultrasound images. In order to account for the opposite contrast between the two domains, i.e., enhancing ultrasound black-colored masses’ translation into light-colored optical images of the masses, the loss term was modified to take the negative value of the input images; i.e.,
(4)$$\begin{array}{cc} L_{opposite}\left(G_{PA},G_{US}\right)= & E_{x∼pdata(x)}\left[|\right|G_{US}\left(\bar{x}\right)-{x||}_{1}]+\\ \; & E_{y∼pdata(y)}\left[|\right|G_{PA}\left(\bar{y}\right)-{y||}_{1}] \end{array}$$
where $\bar{x}$ and $\bar{y}$ are the negative images of *x* and *y* correspondingly.

### 2.2. BUSI Dataset

A public dataset called “Breast Ultrasound Images” (BUSI) collected from women between 25 and 75 years old [27] was used. The data were collected in 2018, from 600 female patients, and consist of 780 images classified to three groups: (i) 133 normal images without masses, (ii) 437 images with benign masses, (iii) 210 images with malignant masses. The images were scanned by a LOGIQ E9 ultrasound system and include additional manually traced masks of the radiologist’s evaluation. The images are in a PNG format, vary in height and width, and have an average size of 600 × 500 pixels. In our study, the data were preprocessed by removing text and labels that were not part of the original ultrasound images and cropping them to partially overlapping square patches of 450 × 450 pixels.

### 2.3. Optic/Anatomic Dataset

To demonstrate feasibility, a set of optical images were collected from poultry breast-tissue samples. To simulate abnormal tissue, small regions were thermally etched. The etched texture resembles the appearance of breast masses (see, for example, [28,29,30,31]). In order to simulate the typical ultrasound signal decay with depth, part of each image contained a black background on the lower part of the image. The etched sizes, shapes, and locations of the simulated tumors were created so as to resemble the relative distribution of the abnormal masses in the BUSI ultrasound images.

### 2.4. Training Parameters

In order to train the model, the dataset, after being cropped into partially overlapping square patches, was divided into train, validation, and test subsets. The training subgroup consisted of 80% of the images, 5% were used for validation, and the rest were used for testing the performance of the suggested method. The relative numbers of cases of malignant, benign, and normal tissue were maintained in the validation and test groups as well. The network’s architecture was based on the CycleGAN architecture provided by [24]. The hyperparameters were tuned according to the validation set, on which the cycle loss was 10× higher than the GAN’s loss and the opposite loss was 0.03 lower. The model was trained on a NVIDIA Tesla V100 GPU (Petach Tikva, Israel) running Linux. Training computation time was approximately 36 h.

### 2.5. Automated Segmentation

In order to evaluate the quality of the generated images, image segmentation was performed and compared to the BUSI’s traced masks. The optical images were segmented by applying the morphological geodesic active contours (MorphGAC) [32] using the scikit-image implementation [33]. As a preprocessing step to highlight the edges, the inverse Gaussian gradient (IGG) was applied. To control the steepness of the inversion, the alpha parameter was set to 100, and the standard deviation of the Gaussian filter sigma parameter was set to 1.5, for both the ultrasonic and the pseudo anatomical display images.

Stemming from the fact that the BUSI tracing was preformed manually, it inherently included inconsistencies. In order to overcome this problem, the generated images were also compared to automatically re-segment masks with the MorphGAC algorithm for the ultrasound images. The code for the segmentation graphical user interface implementation is available https://github.com/LilachBarkat/MorphGAC-Segmentation-GUI (accessed on 27 May 2022).

### 2.6. Evaluation Protocol

Since an image interpretability comparison is ultimately subjective and evaluation metrics such as SSIM (structural similarity) which are commonly used in image reconstruction quality analysis are irrelevant in our case, other indices which are more indicative of the clinical merit were applied. The performance of the method was evaluated by comparing the optical segmentation mask to both the original BUSI reference masks and the BUSI masks re-segmented by the MorphGAC algorithm. Three metrices were used to evaluate the quality of the segmentation results. The first index used for contour evaluation was the Dice index, which assesses shape similarity and is defined as [34]:(5)$$\begin{array}{c} \mathrm{Dice}=\frac{2\times TP}{\left(TP+FP)+(TP+FN\right)} \end{array}$$
where *TP*, *FP*, and *FN* are the true positive, false positive, and false negative pixels, respectively. Positive means within the lesion mask, and negative means outside the mask.

In addition, the accuracy at locating the center of mass of the lesion is defined as:(6)$$\begin{array}{cc} \; & \mathrm{Center}\mathrm{error}=\\ \; & \frac{\sqrt{{\left(C_{x_{Reference}}-C_{x_{Generated}}\right)}^{2}+{\left(C_{y_{Reference}}-C_{y_{Generated}}\right)}^{2}}}{\sqrt{2}a} \end{array}$$
where $C_{x_{Reference}}$ and $C_{x_{Generated}}$ are the *x* coordinates of the center of mass; and $C_{y_{Reference}}$ and $C_{y_{Generated}}$ are the *y* coordinates of the reference masks and the generated masks, respectively. $\sqrt{2}a$ is a normalizing size factor corresponding to the diagonal length of the image (the largest possible dimension in the image).

The area index of the lesion is defined as:(7)$$\begin{array}{c} \mathrm{Area}\mathrm{index}=\frac{|S_{Reference}-S_{Generated}|}{a^{2}} \end{array}$$
where $S_{Reference}$ and $S_{Generated}$ are the segmented lesion areas of the reference and generated masks, and $a^{2}$ is the size normalization to the image area.

## 3. Results

A set of exemplary images translated from ultrasound to colored pseudo anatomical displays is depicted in Figure 2. As can be observed, a pseudo anatomical display provides superior visual discrimination of the lesion, including much clearer border definition and enhanced contrast, especially for the malignant tumors (see Figure 2C,D). Furthermore, the tissue’s texture is more vivid and realistic. Importantly, in most cases, the algorithm manages to overcome the acoustic artifacts commonly appearing in ultrasonic images, such as the acoustic shadow artifacts, as can clearly observed in Figure 2A, acoustic enhancement artifacts of the tumor’s posterior tissue in Figure 2A,B, and the signal decay with depth, at the bottom of the image, as depicted in Figure 2A. These distinctions highlight the superiority of the optical representation over the ultrasound images, particularly by capturing and representing the fine details.

Although the algorithm was successful in most cases, there were instances where its effectiveness was reduced. One of the difficulties encountered was the depiction of erroneous lesions in normal tissues leading to false-positive readings. This was due to the presence of a darker area in comparison to the surrounding tissue, as demonstrated, for example, in Figure 3C. This issue poses a challenge not only only for the algorithm, but also for radiologists who have to use multiple cross-sectional views from various directions to overcome this problem. Additionally, there were few mistranslations of acoustic artifacts, as depicted in Figure 3A.

Nonetheless, part of the discrepancy can presumably be attributed to inconsistency in the BUSI tracing in “difficult to trace” ultrasonic images, as depicted, for example, in Figure 3B. The variability in the manner in which radiologists segment the tumor results in an inconsistent segmentation process. This is caused by factors such as the indistinct contour of the tumor, the subjectivity of manual segmentation, and the diversity in radiologists’ expertise and interpretation. These factors collectively contribute to the observed discrepancy.

This is also demonstrated, for example, in Figure 4 where three substantially different tracings are presented for the same tumor (example taken from “malignant 3–9” in BUSI). As recalled, to overcome the variations stemming from the manual tracing, the ultrasonic images were re-segmented using the MorphGAC algorithm, which yielded better consistency.

The overall estimation of the performance based on the criteria listed above are outlined in Table 2 and are also graphically depicted in Figure 5. As can be observed, the performance is better for benign tumors, this can be attributed to their more regular shapes. Contrary to that, for the malignant tumors, whose geometry is more irregular and whose borders are commonly blurred, it was more difficult for the model and for the MorphGAC algorithm to accurately segment the lesions.

Upon more closely studying the distribution of the quality indices, the area index and the center of mass error were seen to share similar behavior. For the benign tumor, both error indices are close to zero with a narrow distribution. Contrary to that, the distribution for the malignant tumors is wider. Although most of the masks for the generated images have a small detection error, some cases yielded significantly different masks. Examining the Dice-score distribution, for the benign tumors, the re-segmented masks have higher median score of 0.91 compared to 0.85 of the BUSI manually segmented masks. For the malignancies, the distribution is wider and the re-segmentation median score is 0.70, and for the manually segmented BUSI masks, it is 0.58.

## 4. Discussion

Ultrasound is a common and affordable imaging technique that provides real-time imaging without the use of harmful ionizing radiation. It is often used in real-time during surgery, but its images can be challenging to interpret due to their relatively poor quality. The obtained images are characterized by a granulated appearance, a low signal-to-noise ratio, and substantial speckle noise. Furthermore, their quality is highly dependent on the operator’s skills.

The ultrasound probe enables acquisition of cross-sectional images in arbitrary spatial orientation and offers real-time capabilities. Consequently, ultrasound is used in image-guided interventions and surgeries. Nonetheless, interpreting ultrasound images in the anatomical/pathological context can still be difficult for surgeons. This is due to differences in between the black and white images and the appearance of the actual tissue during surgery, and in some of the cases, the indistinct mass margins. These variations pose a challenge in finding and identifying all the abnormal tissue.

To address this problem, we harnessed the great progress in the field of neural networks for image-to-image translation. There are already applications in the medical field for cross-modality translations. In this work, we chose to approach this issue from a new perspective and translated the ultrasound image into an optical image that would resemble the appearance of the actual tissue. However, due to the inherent limitations of producing corresponding paired images of ultrasound and optical images, we chose to employ a CycleGAN model for this task. This type of network is capable of learning each domain separately and is able to produce unique translations for each image.

To translate between two image domains, there must be sufficient similarity between them. To demonstrate the feasibility of our approach, we generated pseudo-anatomical images with similar tumor shapes, sizes, and a black background that mimics the acoustic shadow seen in ultrasound images. To improve the translation results, we have incorporated a constraint into the loss function to convert the black appearance of the tumors in the ultrasound display to white objects in optical images of the tissue. We evaluated the translation by examining the accuracy of tumor location, area, and contours. Our results (Figure 5 and Table 2) show that the translation maintains the geometrical information.

As for the limitations of the suggested method, it is noted that the optical images used for training in this study were based on etched poultry breast-tissue samples which do not necessarily resemble actual cuts observed during surgery. Although the simulated lesions were prepared to resemble the geometrical features of real breast tumors in terms of size and shape, optimal results presumably can be achieved by using actual optical images of anatomical cuts through breast-tumor tissues acquired during surgery. Nonetheless, this study clearly demonstrated the potential that can be achieved by translating ultrasound images into optical images of human tissue during surgery.

In conclusion, in this work, a method for image translation from ultrasound into a pseudo anatomical display was presented. This image-translation method yields a more vivid and realistic tissue-texture presentation which potentially can enable a more straightforward comparison to the real anatomy. Furthermore, as can be observed, the pseudo anatomical images provide superior visual discrimination of the lesions. The borders are well delineated, and the contrast clarity is substantially improved.

The main contribution of this study was to demonstrate the feasibility of this translation, while preserving the anatomical content. It can serve as an ancillary tool by providing a simultaneous depiction of a real-time ultrasound image alongside a pseudo anatomical representation. Although this study may not yield substantial advantages for medical practitioners proficient in breast ultrasound, it renders considerable benefits to those who lack experience in the field. Additionally, this representation can be helpful for medical training and practices. Overall, our proposed method could potentially lead to faster and better diagnosis and improved clinical outcomes.

## Figures and Tables

**Figure 1 bioengineering-10-00388-f001:**
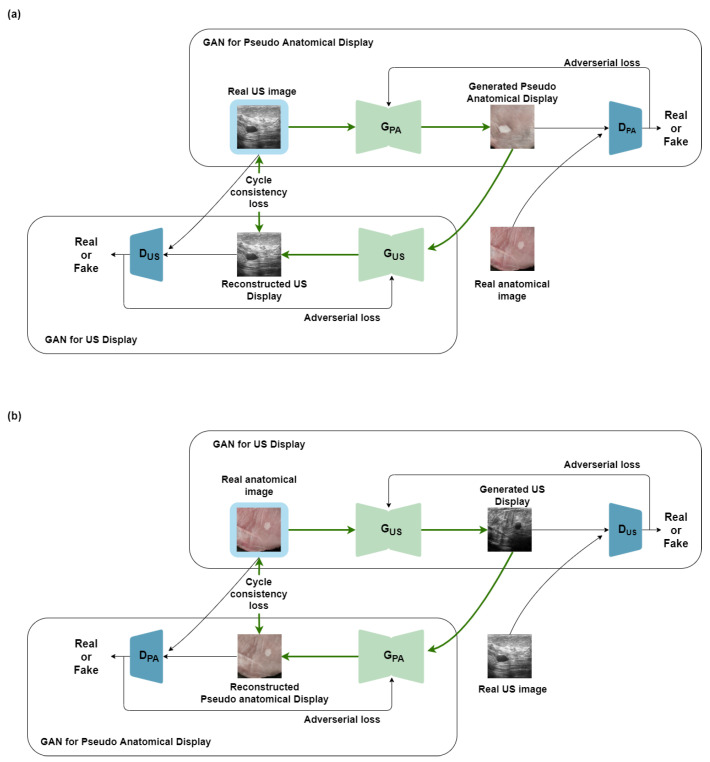
(**a**) Schematic diagram of the CycleGAN model used here to translate real ultrasound images into pseudo anatomical images. The upper block is the GAN for producing the pseudo anatomical display, and the lower block is the GAN for producing the ultrasound display. (**b**) An identical network is used for generating ultrasound displays from anatomical images.

**Figure 2 bioengineering-10-00388-f002:**
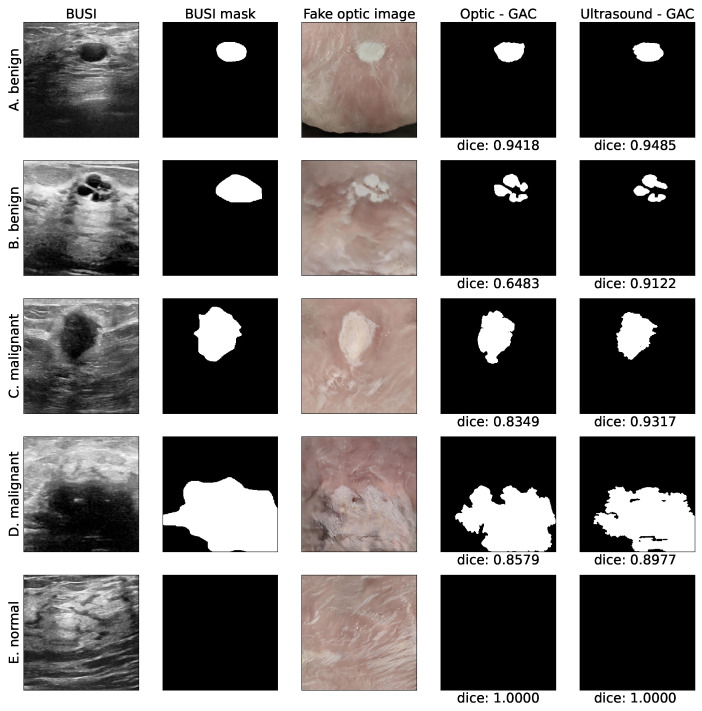
(1st column) Examples of the BUSI images. (2nd column) BUSI masks. (3rd column) The model-generated pseudo anatomical images. (4th column) The corresponding segmented masks obtained by MorphGAC for the pseudo anatomical images and for the original ultrasound images (5th column), with the corresponding Dice scores. The rows correspond to: (**A**) benign, (**B**) benign, (**C**) malignant, (**D**) malignant, and (**E**) normal. As can be noted, the pseudo anatomical images are more natural to comprehend, and the tumors are better defined.

**Figure 3 bioengineering-10-00388-f003:**
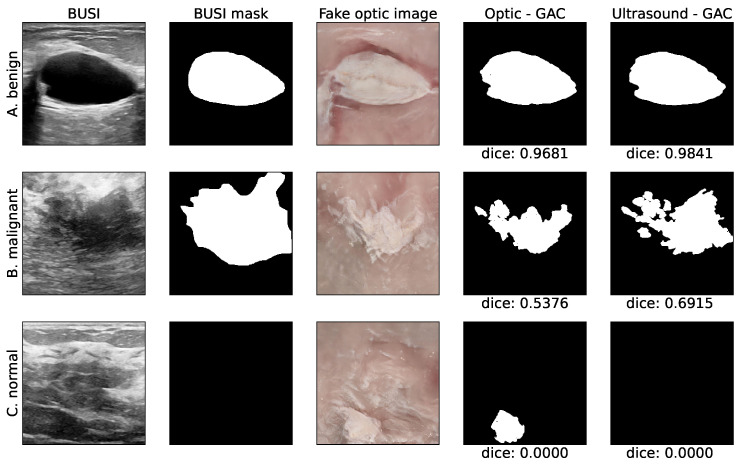
(1st column) Examples of the BUSI images. (2nd column) BUSI masks. (3rd column) The model-generated pseudo anatomical images. (4th column) The corresponding segmented masks obtained by MorphGAC for the pseudo anatomical images and for the original ultrasound images (5th column), with their Dice scores. The rows correspond to: (**A**) benign, (**B**) malignant, (**C**) normal. As can be noted, in these cases, the algorithm was less effective.

**Figure 4 bioengineering-10-00388-f004:**
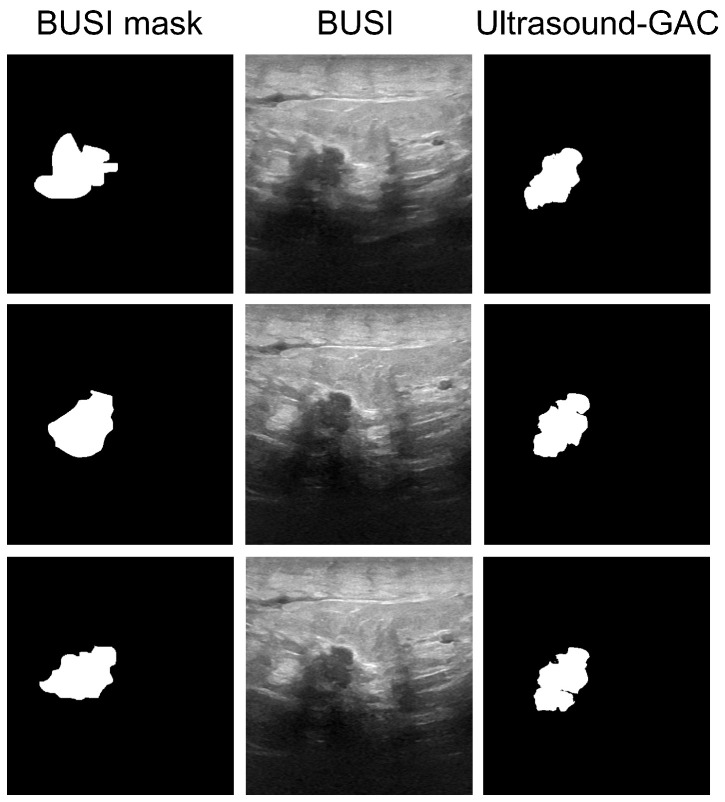
(Middle column) Three exemplary original ultrasonic images of the same tumor, which are marked as “malignant 3–9” in BUSI. (First column) the corresponding three different tracings provided by BUSI for the same tumor. As can be observed, the three tracings differ substantially in shape and geometry. (Last column). Contrary to that, the corresponding MorphGAC segmented masks yielded more consistent tracings, which appear to better match the lesion shapes in the ultrasound images.

**Figure 5 bioengineering-10-00388-f005:**
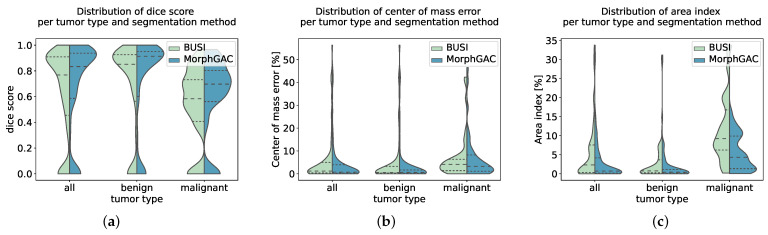
Distributions of the three metrics per tumor type for the two segmentation methods—BUSI (left) MorphGAC (right). (**a**) Dice distribution, (**b**) center error distribution, (**c**) area index distribution. The dashed lines represent the median score and upper and lower quartiles.

**Table 1 bioengineering-10-00388-t001:** Ultrasound imaging pros and cons associated with detection of tumors and their margins, pre-operatively and intra-operatively.

Pros	Cons
Radiation-free	Low SNR
Real time imaging	Operator dependent
Radiation-free	Low SNR
Real time imaging	Operator dependent
Enables arbitrary for cross-section imaging	Non intuitive black and white image
Tumor detection pre and intra-operative	
Margin assessment	
Cost effective	

**Table 2 bioengineering-10-00388-t002:** Quantitative evaluation by comparing the optical segmented mask to both the original BUSI reference masks and the BUSI masks re-segmented by the MorphGAC algorithm.

	Tumor Type	Median BUSI	Median MorphGAC	Mean ± Std BUSI	Mean ± Std MorphGAC
Dice	Benign	0.85	0.91	0.67 ± 0.36	0.70 ± 0.38
	Malignant	0.58	0.70	0.53 ± 0.30	0.60 ± 0.32
	all	0.77	0.83	0.62 ± 0.35	0.67 ± 0.36
Center error [%]	Benign	0.56	0.58	5.09 ± 11.23	4.22 ± 10.78
	Malignant	4.13	3.27	7.21 ± 10.29	7.21 ± 10.65
	all	1.17	0.73	5.76 ± 10.95	5.14 ± 10.79
Area index [%]	Benign	0.74	0.40	2.84 ± 5.21	2.11 ± 5.08
	Malignant	9.25	4.34	11.64 ± 8.79	6.12 ± 6.49
	all	2.31	0.71	5.56 ± 7.67	3.35 ± 5.83

## Data Availability

The data presented in this study are openly available at [27].

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
