# Peer review of "Image Translation of Breast Ultrasound to Pseudo Anatomical Display by CycleGAN"

_bioengineering, 2023, doi:10.3390/bioengineering10030388_

Round 1

Reviewer 1 Report

Dear author

Thank you for the submission of your manuscript. I enjoyed your article.  And I think your paper will be useful to some readers. My comments after reading your paper are as follows:

Angiogenesis and tumor elasticity are important in distinguishing benign and malignant breast tumors by imaging, but the shape of the tumor is the most important. Ultrasonography is an excellent imaging modality that is simple and has no side effects, but its ability to establish images depends greatly on the knowledge and skill of the examiner. The same is true for the interpretation of ultrasound findings. Although this study offers little benefit to medical practitioners familiar with breast ultrasound, it is of great benefit to novices. Furthermore, as clearly shown in Figure 4, it is also wonderful that the final image does not differ significantly even though the initial traces are different. This research is judged to be of great benefit to the readers.

Author Response

Dear Reviewer,

Thank you for taking the time to review our manuscript and for your positive feedback. We appreciate your comments, which have helped us to improve the quality of our work.

We have emphasized in both, the introduction and conclusions sections of the manuscript, the dependency on the knowledge and skill of the examiner and the benefit that our proposed method offers to inexperienced ultrasound operators.

Once again, we appreciate your feedback.

Reviewer 2 Report

The article is well written. It still requires few minor changes as I suggest them below :

Explain the literature section in a comparative manner. Add merits and demerits in the tabular form for better explanation for selected study.

Discuss the real time/ practical application of the paper.

Add major contribution in the paper. What is the significance of this study.

Add Materials and Methods section after Literature review.

Author Response

Dear Reviewer,

Thank you for taking the time to review our manuscript and for your constructive feedback. We appreciate your comments, which helped us to improve the quality of our work.

We have made the suggested changes to the literature section of the manuscript and have included a comparative analysis that highlights the merits and demerits of ultrasound in a tabular form, which we believe will better explain the selected study.

We have changed the Methods section into Materials and Methods and added a discussion on the real time application according to the network inference time.

Finally, we have added in the discussion section a paragraph on the major contribution of our study and its significance. We have highlighted the practical application of the study.

We would like to thank you for your valuable feedback and for guiding us towards making these changes.

Reviewer 3 Report

Thank you for submitting interesting manuscript.

The pseudo-anatomical display by CycleGAN would be very helpful for the medical training and practices.

The ultrasound image is acquired in real-time.

The Figure 2 and Figure 3 are great, but related supplementary videos would be very effective to show the excellency of the study.

Existing human data of ultrasound video might be applied to the algorithm of the study.

Author Response

Dear Reviewer,

Thank you for taking the time to review our manuscript and for your positive feedback. We appreciate your constructive comments.

We added in the introduction and also in the discussion sections the potential of utilizing the pseudo-anatomical display, particularly in the context of medical training and practice.

We appreciate your suggestion for the addition of supplementary ultrasound video of human data. Unfortunately, we are unable to produce a video focused on human breast tumor as we do not have access to the necessary resources and data.

Nonetheless, we added in the methods section a paragraph about the short network inference time and therefore the real time application.

Once again, we appreciate your feedback and will take it into consideration for future research.